# Psychological Determinants of Vaccination Readiness against COVID-19 and Seasonal Influenza of the Chronically Ill in Primary Care in Germany—A Cross-Sectional Survey

**DOI:** 10.3390/vaccines11121795

**Published:** 2023-11-30

**Authors:** Linda Sanftenberg, Simon Keppeler, Nadine Heithorst, Tobias Dreischulte, Marco Roos, Philipp Sckopke, Markus Bühner, Jochen Gensichen

**Affiliations:** 1Institute of General Practice and Family Medicine, University Hospital, LMU Munich, 80336 Munich, Germany; simon.keppeler@med.uni-muenchen.de (S.K.); nadine.heithorst@med.uni-muenchen.de (N.H.); tobias.dreischulte@med.uni-muenchen.de (T.D.); jochen.gensichen@med.uni-muenchen.de (J.G.); 2General Practice, Medical Faculty, University of Augsburg, 86356 Neusäß, Germany; marco.roos@med.uni-augsburg.de; 3Department of Psychology, LMU Munich, 80802 Munich, Germany; philipp.sckopke@psy.lmu.de (P.S.); markus.buehner@psy.lmu.de (M.B.)

**Keywords:** vaccination readiness, chronical illness, depression, anxiety disorder, mental health, psychological antecedents of vaccination, primary care

## Abstract

Vaccines against COVID-19 and influenza are highly recommended for the chronically ill. They often suffer from co-morbid mental health issues. This cross-sectional observational study analyzes the associations between depression (PHQ-9) and anxiety (OASIS) with vaccination readiness (5C) against COVID-19 and influenza in chronically ill adults in primary care in Germany. Sociodemographic data, social activity (LSNS), patient activation measure (PAM), and the doctor/patient relationship (PRA) are examined as well. Descriptive statistics and linear mixed-effects regression models are calculated. We compare data from *n* = 795 study participants. The symptoms of depression are negatively associated with confidence in COVID-19 vaccines (*p* = 0.010) and positively associated with constraints to get vaccinated against COVID-19 (*p* = 0.041). There are no significant associations between symptoms of depression and vaccination readiness against influenza. Self-reported symptoms of a generalized anxiety disorder seem not to be associated with vaccination readiness. To address confidence in COVID-19 vaccines among the chronically ill, targeted educational interventions should be elaborated to consider mental health issues like depression. As general practitioners play a key role in the development of a good doctor/patient relationship, they should be trained in patient-centered communication. Furthermore, a standardized implementation of digital vaccination management systems might improve immunization rates in primary care.

## 1. Introduction

Chronic physical illnesses account for 41 million deaths worldwide every year, e.g., coronary artery disease (CAD; 17.9 million deaths annually), followed by cancers (9.3 million), respiratory diseases (bronchial asthma, chronic obstructive pulmonary disease (COPD); 4.1 million), and diabetes (2.0 million), according to a World Health Organization report from 2023 [1]. Chronic illnesses put patients at risk for increased morbidity and mortality from infectious diseases such as COVID-19 and seasonal influenza [2,3]. Therefore, the chronically ill and adults with significant comorbidities are priority groups for vaccinations against COVID-19 and influenza [4,5,6]. To slow down the spread of the virus, researchers around the world developed different types of vaccines. New-generation mRNA vaccines have been available since 2021 in addition to conventional viral vector vaccines, showing excellent data on immunogenicity, safety, and effectiveness [7]. However, depression and anxiety disorders are closely associated with chronic physical illness and worsen the health outcomes of those patients [8,9].

According to a case-control study using electronic health records representing 20% of the US population, patients with depression become infected more often with COVID-19, are more likely to be hospitalized (27.4% vs. 18.6%, *p* < 0.001), and have higher mortality due to COVID-19 (8.5% vs. 4.7%, *p* < 0.001) in comparison with the general public [8,10]. Consequently, vaccinations against COVID-19 are also recommended for patients with chronic mental illnesses such as anxiety and depression in different healthcare systems [3,11,12]. Unfortunately, COVID-19 vaccine hesitancy has increased in people suffering from depressive symptoms [13,14,15].

So far, it cannot be determined if people with depressive symptoms and anxiety disorders face an increased risk of an influenza infection [16]. However, there is a negative impact of depression on vaccination behavior against influenza [17,18]. Depression has a negative effect on the utilization of preventive health services and is a risk factor for non-adherence to medical treatment [19]. However, the psychological antecedents of (non-)vaccination in the chronically ill are still unclear. These insights are of utmost importance to elaborate target-group specific interventions and improve clinical guidelines as well as health care campaigns for the chronically ill in primary care. The 5C model can be used to measure vaccination readiness and describes five relevant psychological antecedents of vaccination: confidence (trust in efficacy and safety of vaccines as well as the corresponding health care system), complacency (risk perceptions), constraints (barriers), calculation (extent of information search and evaluation), and collective responsibility (willingness to protect the community) [20].

There are additional patient-dependent variables that might mediate the psychological antecedents of vaccination readiness. These are mainly social activity, patient activation measures, and the doctor/patient relationship. Social networks and interpersonal relationships play a major role in disseminating information about vaccination, as well as supporting the vaccination decision-making process [21,22]. Sufficient patient activation can be understood as a basic requirement for health-related self-management accompanied by changes in health behavior. Furthermore, individuals with high levels of patient activation measures show positive attitudes toward preventive behaviors, including immunizations [23]. A good doctor/patient relationship has a strong influence on vaccination rates, as patients primarily obtain information about vaccinations in primary care and usually follow the respective recommendations [24,25,26].

As primary care plays a key role in the diagnosis, treatment, and vaccination of the chronically ill, the aim of this study is to analyze the associations between depression, anxiety disorder, and vaccination readiness in German primary care patients with at least one chronic physical illness.

## 2. Materials and Methods

### 2.1. Data Collection

Eligible patients were identified in cooperation with 13 general practices in Germany. A purposive sample of general practices was invited for participation (five solo practices, six group practices, and two medical care centers, from both urban and rural regions). Using electronic patient management systems, we identified patients, who were at least 18 years of age and visited their general practice within the last six months. Furthermore, for inclusion, there had to be at least one diagnosed chronic physical illness: bronchial asthma, COPD, diabetes type 1 or 2, CAD, or breast cancer. A total of 3152 patients meeting those criteria were contacted by their general practice and invited to answer a paper-based questionnaire by mail between August and October 2022. Questionnaires that were sent back until 1 December 2022 were included in this study.

### 2.2. Instruments

For sociodemographic characteristics, study participants were asked for their age, sex, education, and their living situation (living alone/living with others).

Vaccination readiness was measured using the validated German version of the 5C model, asking about influenza and COVID-19 separately [20]. Study participants answer with a Likert scale from 1 (“I strongly disagree”) to 7 (“I strongly agree”). The questionnaire is evaluated at the item level. High levels of agreement with the items “confidence” and low levels of agreement with the items “complacency”, “constraints”, “calculation”, and “collective responsibility” are associated with an increased vaccination readiness [20].

Symptoms of depression were measured using the Patient Health Questionnaire-9 (PHQ-9) score [27]. This is a validated self-administered questionnaire consisting of nine items, each scoring one of the Diagnostic and Statistical Manual of Mental Disorders IV (DSM-IV) criteria for major depression with a sum score ranging from 0 to 27. A sum score of at least 10 indicates major depression. The items assess symptoms within the last two weeks with a Likert scale from 0 (“not at all”) to 4 (“almost every day”). Sensitivity is reported to be 0.80 (95% CI [0.71, 0.87]) and specificity to be 0.92 (95% CI [0.88, 0.95]) with a cut-off of 10 or higher.

Symptoms of a general anxiety disorder were measured using the Overall Anxiety Severity and Impairment Scale (OASIS) score [28]. This validated self-administered questionnaire consists of five items and measures anxiety severity and impairment in daily activities with a sum score ranging from 0 to 20. A sum score of at least 8 indicates a clinically relevant anxiety disorder. Participants answer using a Likert scale from 0 (“no/none”) to 4 (“extreme/frequent/all the time”).

The Lubben Social Network Scale (LSNS) consists of six items in two subscales (family and friends) and assesses data about social activity [29]. A sum score between 0 and 30 is calculated, where higher scores indicate higher levels of social activity and stronger social networks. Values below 11 indicate a high risk for social isolation.

The Patient Activation Measure 13 (PAM13) consists of 13 items and assesses data on patients’ active participation in their medical care and self-management [30]. The sum score ranges between 13 and 52, with higher values indicating higher patient activation.

The Patient Reactions Assessment (PRA) consists of 15 items in three subscales (information, affectivity, communication) and measures the subjectively perceived quality of the doctor/patient relationship [31]. The sum score ranges between 15 and 105, with higher values indicating a better doctor/patient relationship.

### 2.3. Data Analysis

Received questionnaires were excluded from the data analysis if less than a third of the total questionnaire was answered. Missing data concerning the patient activation measure (PAM) and the doctor/patient relationship (PRA) were calculated according to the suggested imputation procedures for both instruments [30,31].

Descriptive statistics were calculated as means and standard deviations, respectively, frequencies and percentages. To assess the association between mental health and vaccination readiness, we calculated Spearman’s rank correlation coefficient. Linear mixed-effects models were chosen for regression since the patients were clustered into 13 different practices with potential effects on vaccination readiness. A total of ten models were calculated, each with one of the 5C items as the dependent variable for both COVID-19 and influenza. They were regressed against the sum scores of the PHQ-9, OASIS, LSNS, PAM13, and PRA, further adjusting for age, sex, education, and living situation. Rural or urban residency of the study participants was not considered since it was assessed indirectly using the GP’s residency and thus controlled for in the mixed-effect models. Statistical significance was based on a *p*-value of 0.05. All analyses were carried out in SPSS 28 and R Version 4.2.2

## 3. Results

### 3.1. Sociodemographic Characteristics

The response rate was *n* = 864 (27.4%). After the exclusion of *n* = 46 questionnaires that were filled out invalidly (less than one-third of the total questionnaire was answered), the total sample size was *n* = 795. The sociodemographic baseline characteristics of the study participants are shown in Table 1. As we included incomplete questionnaires in our analysis, missing data on the item level are indicated. The mean age of the patients was 67 years (18–94 years). Men and women were equally represented (47.4% female patients). Most patients had lower education (without university qualification, 58.6%) and lived together with others (73.5%).

### 3.2. Psychological Determinants of Vaccination Readiness

Figure 1 illustrates mean scores and standard deviations of the examined psychological determinants of vaccination readiness against COVID-19 and influenza. On average, the study participants show high levels of confidence in the effectiveness and safety of vaccinations against COVID-19 and influenza, the health system, and policymakers’ motivation (item “confidence” in COVID-19 vaccination: M = 5.67, SD = 1.38, item “confidence” in influenza vaccination: M = 5.65, SD = 1.41). Furthermore, they seem to search extensively for information about vaccinations against COVID-19 and influenza and try to evaluate them (item “calculation” COVID-19 vaccination: M = 5.19, SD = 1.90, influenza vaccination: M = 5.12, SD = 1.83).

Perceived risks concerning an infection with COVID-19 and influenza seem to be moderate, and vaccinations are considered necessary preventive measures (item “complacency” toward COVID-19: M = 1.66, SD = 1.21, item “complacency” toward influenza: M = 2.03, SD = 1.34). The physical availability, affordability, and accessibility of both vaccinations are not perceived as actual constraints the getting immunized (item “constraints” COVID-19: M = 1.37, SD = 1.00; item “constraints” influenza: M = 1.61, SD = 1.15). Furthermore, our study participants are willing to protect others with their own vaccination by means of herd immunity (item “collective responsibility” COVID-19: M = 1.51, SD = 1.04; item “collective responsibility” influenza: M = 1.67, SD = 1.18).

To sum up, psychological antecedents concerning vaccinations against COVID-19 and influenza are comparable, with no significant differences. However, an infection with COVID-19 was perceived to be a bit more risky than an infection with influenza, and the study participants perceived a few more organizational constraints to getting immunized against COVID-19 in comparison with influenza.

Symptoms of major depression were reported by *n* = 146 of our study participants (18.4%). N = 161 of our study participants (20.3%) showed symptoms of a clinically relevant anxiety disorder.

Spearman’s rank correlation revealed a strong positive correlation between depression and anxiety in this sample (r = 0.60, *p* < 0.01). The mean LSNS sum score of 13.59 indicated a widespread risk of social isolation in our study population (see Table 2).

### 3.3. Linear Mixed Regression Models

Mixed linear regression models regarding the psychological antecedents for vaccinations against COVID-19 showed that “confidence” was negatively associated with depression (β = −0.04, 95% CI [−0.07, −0.01], *p* ≤ 0.010). “Confidence” was positively associated with higher patient activation measure (PAM13, β = 0.03 95% CI [0.01, 0.05], *p* = 0.005), a subjectively perceived high quality of a doctor/patient relationship (PRA, β = 0.02, 95% CI [0.01, 0.03]), *p* = 0.003), and older age (β = 0.02 95% CI [0.01, −0.03]), *p* ≤ 0.001). “Complacency” was negatively associated with a good doctor/patient relationship (PRA, β = −0.02, 95% CI [−0.03, −0.01], *p* ≤ 0.001), age (β = −0.01, 95% CI [−0.01, −0.00], *p* = 0.013), and education (β = −0.22, 95% CI [−0.41, −0.02], *p* = 0.027). “Constraints” were positively associated with depression (PHQ9, β = 0.02, 95% CI [0.001; 0.04], *p* = 0.041) and negatively associated with lower education (β = −0.25, 95% CI [−0.10, −0.04], *p* ≤ 0.001]. “Calculation” was positively associated with patient activation (PAM13, β = 0.04, 95% CI [0.01, −0.07], *p* = 0.018). Lastly, “collective responsibility” was negatively associated with a good doctor/patient relationship (PRA, β = −0.02, 95% CI [−0.02, −0.01], *p* ≤ 0.001; see Table 3 and Table 4).

Mixed linear regression models regarding the psychological antecedents for vaccinations against influenza showed that “confidence” was positively associated with patient activation (PAM13, β = 0.03 95% CI [0.00, 0.05], *p* = 0.024), the doctor/patient relationship (PRA, β = 0.02, 95% CI [0.01, 0.03]), *p* ≤ 0.001), and older age (β = 0.02 95% CI [0.01, 0.03]), *p* ≤ 0.001). “Complacency” was negatively associated with patient activation (PAM13, β = −0.03, 95% CI [−0.05, −0.01], *p* = 0.001), a subjectively perceived high-quality doctor/patient relationship (PRA, β = −0.02, 95% CI [−0.03, −0.01], *p* ≤ 0.001), and older age (β = −0.02, 95% CI [−0.02, −0.01], *p* ≤ 0.001). “Constraints” were positively associated with a subjectively perceived high-quality doctor/patient relationship (PRA, β = −0.01, 95% CI [−0.02, −0.001]), *p* = 0.070) and older age (β = −0.01, 95% CI [−0.01, −0.001]), *p* = 0.024) and negatively associated with education (β = −0.20, 95% CI [−0.39, −0.03]), *p* = −0.24). “Calculation” was positively associated with patient activation (PAM13, β = 0.03, 95% CI [0.001, 0.06], *p* = 0.042) and living situation (β = 0.42, 95% CI [0.06, 0.77]), *p* = 0.023). Finally, higher “Collective Responsibility” was positively associated with a subjectively perceived high-quality doctor/patient relationship (PRA, β = −0.02, 95% CI [−0.03, −0.01], *p* ≤ 0.001) and the female sex (β = −0.20, 95% CI [−0.38, −0.02], *p* = 0.028; see Table 5 and Table 6).

## 4. Discussion

Within our cross-sectional survey, we could show that (1) depression was negatively associated with confidence in the safety and efficacy of vaccinations against COVID-19 and the respective healthcare system. Furthermore, (2) depression led to a subjectively perceived increase in constraints on vaccinations against COVID-19. Beyond mental health, we determined patient activation measures and the perceived quality of the doctor/patient relationship as important patient-dependent variables affecting vaccination readiness in German primary care.

A US study identified “confidence” as the most important psychological antecedent explaining COVID-19 vaccine hesitancy in the general public [32]. Targeted and tailored public health interventions that enhance the public’s confidence in vaccines and emphasize the risk and seriousness of COVID-19 may address COVID-19 vaccine hesitancy. Authoritative figures not linked to vaccine competence (such as religious leaders) could be involved in communication campaigns [33], and unconventional communication channels like social media campaigns could be used to reach predefined groups of special interests (such as self-help groups for certain chronic diseases) [34]. Furthermore, vaccination campaigns should take advantage of the public attention that the COVID-19 pandemic has brought to vaccination medicine and integrate the psychological determinants of vaccination readiness into their strategy [35]. The public knowledge attitude regarding public health basics (such as herd immunity) should be used to increase the vaccination rates of other standard and indication vaccinations in addition to regular COVID-19 vaccinations.

To promote vaccine confidence in vulnerable groups like the chronically ill suffering from psychological co-morbidities, special efforts are needed. A suitable measure might be targeted educational interventions like communication-based short interventions based upon shared decision-making [25,36]. This kind of patient-centered and bidirectional doctor–patient communication style accounts for medical as well as personal aspects of the individual patient and is, therefore, suitable to strengthen confidence in the safety and effectiveness of vaccines [37,38].

Additionally, our results show that depression led to a subjectively higher perception of “constraints” to getting immunized against COVID-19. Comparable to the general population, this perception may arise from negative perceptions or misinformation about vaccine safety and efficacy [39]. However, this distrust may be amplified by experiences of stigma from healthcare providers and experienced socioeconomic as well as structural barriers within the healthcare system [40].

This has been shown to be associated with the perception of decreased self-efficacy, which can undermine follow-through with vaccination [13,41]. Our results are in line with previous studies that identified lower (perceived) self-efficacy as well as increased levels of vaccine hesitancy against COVID-19 vaccinations in depressed adult patients suffering from chronic conditions [13,42,43]. A meta-analysis reported a threefold decrease in compliance among depressed patients with respect to their recommended medical treatment. Depression might interfere with patients’ adherence as they show attitudes of hopelessness and pessimism, which can make active actions seem worthless to the patient. Second, depression is associated with social isolation and thus withdrawal from individuals who could provide support, assistance, and information. Third, depression may affect cognitive function, which makes it difficult to remember and carry out actions [19]. These aspects are assumed to negatively affect vaccination rates against COVID-19. In a sample of non-institutionalized patients with mental health conditions in Denmark, 84.8% reported that they were willing to be vaccinated for COVID-19 compared with 89.5% of the general population [39].

To tackle subjectively perceived “constraints”, reminder systems are an easy-to-implement measure, as they provide up-to-date information and regular reminders about pending immunizations via text messages for the primary care team and the individual patient [44,45].

Study participants with symptoms of anxiety did not differ in their psychological antecedents to getting vaccinated against COVID-19 in our study. These observations are in line with international analyses, that measured possible relationships between vaccine hesitancy and intolerance of uncertainty, COVID-19 stress, disgust, and time spent researching information about vaccination in people high in anxiety symptoms [44].

Concerning influenza, we did not identify an association between mental health and vaccination readiness. This observation is in line with another German study concerning depression and vaccination behavior against influenza in adults with COPD [46]. Another German study concluded that the broader construct of anxiety showed no association with vaccine acceptance in a German convenience sample. They further state that instead of general anxiety, specific fears and anxiety (e.g., fear of getting infectious diseases like COVID-19 or influenza, fear of social and economic consequences) have associations with vaccine acceptance in both directions [47]. This might explain why the present study did not find associations since we only assessed anxiety in general. So, general anxiety on an individual level can have bidirectional effects that are not detectable with a regression model.

As already shown, a subjectively perceived high-quality doctor/patient relationship seems to have a strong positive association with vaccination readiness [24,25]. Furthermore, we confirmed that older age, the male sex, and higher education are associated with increased vaccination readiness in patients with chronic conditions [17,46,48]. The role of patient activation was ambivalent, being associated with higher scores for “confidence” and “calculation”. As “calculation” measures extensive information searching and evaluation, “calculation” per se is not a predictor of reduced vaccination readiness. However, sufficient searching for and evaluating health-related information requires a certain level of health literacy. Otherwise, it can be assumed that there will be a negative impact on vaccination readiness [20]. Further research is needed to examine the effect of health literacy on vaccination readiness in German chronically ill with and without mental health issues.

With our survey, we were able to reach chronically ill primary care patients who are not easy to address using digital survey tools. We could include a broad range of ages (18–94 years) and a high level of heterogeneity in terms of rural/urban residency. This might increase the external validity of our findings.

However, we have to consider a selection bias, as our survey was obviously about vaccines, and participating practices and patients might have a dedicated attitude toward vaccinations. We only invited patients with a diagnosed chronic illness who were well-integrated into primary care, had a good doctor/patient relationship, and had the sufficient ability, opportunity, and motivation to take care of their own health. To examine the proportion of variance that is attributable to respective general practices, an intra-cluster correlation would be necessary. At the same time, depression, lack of self-management skills, and patient-activation measures might have discouraged potential study participants from active participation. As we analyzed only patient-reported outcomes, response bias might also be relevant. Furthermore, vaccination readiness might change during this ongoing dynamic pandemic, and the observed effect sizes are relatively small.

## 5. Conclusions

Patients suffering from chronic conditions might benefit from trustful patient-centered communication about sufficient preventive measures, especially if they suffer from mental health issues. General practitioners play a key role in the care of these patients and in the long-term development of a mutually trustful relationship. Consequently, general practice teams should receive regular training in appropriate patient-centered communication techniques, following the principles of shared decision-making. Furthermore, the standardized implementation of digital vaccination management system would support general practice teams as well as their patients to improve vaccination rates in primary care.

## Figures and Tables

**Figure 1 vaccines-11-01795-f001:**
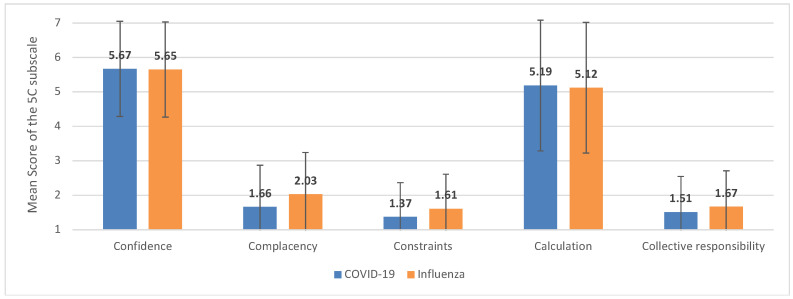
Mean scores of the psychological antecedents of vaccination for the COVID-19 vaccine and the influenza vaccine (*n* = 774).

**Table 1 vaccines-11-01795-t001:** Baseline characteristics of the study participants (*n* = 795).

Variable	Categories	Values
Age, M (SD)	-	67 (14)
Biological sex, *n* (%)	Male	417 (52.5%)
	Female	377 (47.4%)
	Diverse	1 (0.1%)
	Missing data	0 (0.0%)
Education, *n* (%)	Lower level of education (without university qualification)	466 (58.6%)
	Higher level of education (with university qualification)	275 (34.6%)
	Missing data	54 (6.8%)
Living situation, *n* (%)	Living alone	184 (23.1%)
	Living with others	584 (73.5%)
	Missing data	27 (3.4%)

**Table 2 vaccines-11-01795-t002:** Results of the validated instruments included in the survey.

Instrument (Topic of Interest, Range of Sum Score)	Mean Sum Score (SD)	Missing Values (%)	Cronbach’s Alpha
PHQ-9 (depression, 0–27)	5.96 (4.54)	22 (2.8%)	0.83
OASIS (generalized anxiety, 0–20)	4.02 (4.45)	14 (1.8%)	0.93
LSNS (social activity, 0–30)	13.59 (5.30)	5 (0.6%)	0.84
PAM13 (patient activation, 13–52)	43.46 (5.92)	6 (0.8%)	0.86
PRA (doctor/patient relationship, 15–105)	91.95 (11.31)	126 (15.8%)	0.90

**Table 3 vaccines-11-01795-t003:** Mixed linear regression models regarding the psychological antecedents for vaccinations against COVID-19.

Psychological Antecedents	Predictor	β	95% CI	*t*-Value	df	*p*-Value
Confidence (*n* = 609)	Intercept	5.71	[5.39, 6.03]	35.06	51.18	<0.001
	PHQ-9 sum	−0.04	[−0.07, −0.01]	−2.57	593.21	0.010
	OASIS sum	0.01	[−0.02, 0.04]	0.94	593.63	0.345
	LSNS sum	0.01	[−0.01, 0.03]	0.73	593.42	0.463
	PAM13 sum	0.03	[0.01, 0.05]	2.81	596.00	0.005
	PRA sum	0.02	[0.01, 0.03]	2.97	596.96	0.003
	Age	0.02	[0.01, 0.03]	4.41	598.91	<0.001
	Sex = female	−0.15	[−0.37, 0.06]	−1.42	594.95	0.156
	Education = higher education	0.15	[−0.07, 0.37]	1.32	598.27	0.189
	Living situation = living with others	−0.08	[−0.33, 0.17]	−0.64	599.00	0.520
Complacency (*n* = 609)	Intercept	1.83	[1.54, 2.12]	12.16	41.17	<0.001
	PHQ-9 sum	0.01	[−0.02, 0.03]	0.51	592.60	0.610
	OASIS sum	−0.00	[−0.03, 0.02]	−0.06	592.70	0.951
	LSNS sum	−0.01	[−0.03. 0.01]	−1.05	592.70	0.2962
	PAM13 sum	−0.01	[−0.03, 0.00]	−1.55	594.80	0.123
	PRA sum	−0.02	[−0.03, −0.01]	−4.75	595.40	<0.001
	Age	−0.01	[−0.02, −0.00]	−2.49	598.60	0.013
	Sex = female	−0.04	[−0.22, 0.14]	−0.45	593.80	0.657
	Education = higher education	−0.22	[−0.40, −0.03]	−2.21	598.80	0.027
	Living situation = living with others	−0.10	[−0.32, 0.11]	−0.93	598.50	0.354
Constraints (*n* = 607)	Intercept	1.546	[1.36, 1.73]	16.05	597.00	<0.001
	PHQ-9 sum	0.02	[0.00, 0.04]	2.05	597.00	0.041
	OASIS sum	−0.01	[−0.03, 0.01]	−0.73	597.00	0.468
	LSNS sum	−0.00	[−0.02, 0.01]	−0.59	597.00	0.556
	PAM13 sum	0.01	[−0.01, 0.02]	0.64	597.00	0.520
	PRA sum	−0.01	[−0.01, 0.00]	−1.71	597.00	0.088
	Age	−0.00	[−0.01, 0.01]	−0.02	597.00	0.986
	Sex = female	−0.05	[−0.20, 0.11]	−0.60	597.00	0.551
	Education = higher education	−0.25	[−0.41, −0.10]	−3.25	597.00	0.001
	Living situation = living with others	−0.15	[−0.33, 0.02]	−1.70	597.00	0.091
Calculation (*n* = 607)	Intercept	4.98	[4.59, 5.36]	25.05	597.00	<0.001
	PHQ-9 sum	0.01	[−0.03, 0.06]	0.67	597.00	0.504
	OASIS sum	0.00	[−0.04, 0.04]	0.08	597.00	0.935
	LSNS sum	0.01	[−0.02, 0.04]	0.71	597.00	0.480
	PAM13 sum	0.04	[0.01, 0.07]	2.38	597.00	0.018
	PRA sum	0.01	[−0.00, 0.03]	1.78	597.00	0.076
	Age	−0.00	[−0.02, 0.01]	−0.68	597.00	0.495
	Sex = female	0.06	[−0.25, 0.37]	0.36	597.00	0.718
	Education = higher education	−0.01	[−0.32, 0.31]	−0.06	597.00	0.952
	Living situation = living with others	0.31	[−0.05, 0.67]	1.67	597.00	0.094
Collective responsibility (*n* = 609)	Intercept	1.51	[1.31, 1.70]	14.90	129.72	<0.001
	PHQ-9 sum	0.01	[−0.02, 0.03]	0.55	596.75	0.580
	OASIS sum	0.01	[−0.02, 0.03]	0.52	598.00	0.602
	LSNS sum	−0.01	[−0.02, 0.00]	−1.32	598.59	0.188
	PAM13 sum	−0.00	[−0.02, 0.01]	−0.33	598.88	0.741
	PRA sum	−0.02	[−0.02, −0.01]	−4.39	589.41	<0.001
	Age	−0.00	[−0.01, 0.00]	−0.87	561.98	0.387
	Sex = female	−0.13	[−0.28, 0.02]	−1.71	599.00	0.089
	Education = higher education	−0.11	[−0.27, 0.04]	−1.43	539.13	0.153
	Living situation = living with others	0.05	[−0.13, 0.23]	0.52	588.96	0.601

**Table 4 vaccines-11-01795-t004:** Significant associations between patient-dependent variables and psychological antecedents for vaccinations against COVID-19. ells representing significant associations are highlighted in orange. Those associations relevant to both COVID-19 and influenza are highlighted in a dark orange. (**p* ≤ 0.05, **: *p* ≤ 0.01, ***: *p* ≤ 0.001).

COVID-19 Vaccine	Confidence	Complacency	Constraints	Calculation	Collect. Respons.
Depression (PHQ9)	**−0.04 (*p* = 0.010)		*0.02 (*p* = 0.041)		
Patient-activation (PAM13)	**0.03 (*p* = 0.005)			*0.04 (*p* = 0.018)	
Doctor/patient relationship (PRA)	**0.02 (*p* = 0.003)	***−0.02 (*p* < 0.001)			***−0.02 (*p* < 0.001)
Age	***0.02 (*p* ≤ 0.001)	*−0.01 (*p* = 0.013)			
Higher education		*−0.22 (*p* = 0.027)	***−0.25 (*p* = 0.001)		

**Table 5 vaccines-11-01795-t005:** Mixed linear regression models regarding the psychological antecedents for vaccinations against influenza.

Psychological Antecedents	Predictor	β	95% CI	*t*-Value	df	*p*-Value
Confidence (*n* = 608)	Intercept	5.53	[5.21, 5.85]	33.56	80.79	<0.001
	PHQ-9 sum	−0.02	[−0.05, 0.01]	−1.32	594.26	0.189
	OASIS sum	0.00	[−0.03, 0.03]	0.21	594.72	0.837
	LSNS sum	0.01	[−0.01, 0.03]	0.75	594.20	0.456
	PAM13 sum	0.03	[0.00, 0.05]	2.26	596.50	0.024
	PRA sum	0.02	[0.01, 0.03]	3.83	597.32	<0.001
	Age	0.02	[0.01, 0.03]	4.15	597.17	<0.001
	Sex = female	−0.00	[−0.23, 0.22]	−0.04	595.73	0.970
	Education = higher education	0.08	[−0.15, 0.31]	0.66	595.53	0.510
	Living situation = living with others	0.09	[−0.18, 0.35]	0.65	597.82	0.516
Complacency (*n* = 608)	Intercept	2.00	[1.72, 2.28]	13.831	79.08	<0.001
	PHQ-9 sum	0.01	[−0.02, 0.04]	0.71	594.11	0.478
	OASIS sum	−0.00	[−0.03, 0.02]	−0.19	594.64	0.853
	LSNS sum	−0.01	[−0.03, 0.01]	−1.16	594.10	0.247
	PAM13 sum	−0.03	[−0.05, −0.01]	−3.30	596.54	0.001
	PRA sum	−0.02	[−0.03, −0.01]	−3.59	597.42	<0.001
	Age	−0.02	[−0.02, −0.01]	−4.03	596.83	<0.001
	Sex = female	−0.14	[−0.34, 0.05]	−1.43	595.74	0.153
	Education = higher education	0.00	[−0.20, 0.20]	0.03	594.70	0.974
	Living situation = living with others	0.04	[−0.19, 0.28]	0.37	597.71	0.710
Constraints (*n* = 606)	Intercept	1.65	[1.42, 1.86]	14.54	175.20	<0.001
	PHQ-9 sum	0.00	[−0.02, 0.02]	0.06	594.70	0.950
	OASIS sum	−0.00	[−0.03, 0.02]	−0.27	595.60	0.787
	LSNS sum	−0.01	[−0.03, 0.01]	−1.26	596.00	0.210
	PAM13 sum	−0.02	[−0.03, 0.00]	−1.81	594.90	0.070
	PRA sum	−0.01	[−0.02, −0.00]	−2.38	576.60	0.018
	Age	−0.01	[−0.01, 0.01]	−2.26	547.00	0.024
	Sex = female	−0.06	[−0.23, 0.11]	−0.66	597.00	0.510
	Education = higher education	−0.20	[−0.39, −0.03]	−2.26	528.00	0.024
	Living situation = living with others	−0.03	[−0.23, 0.17]	−0.33	585.10	0.744
Calculation (*n* = 605)	Intercept	4.78	[4.40, 5.15]	24.49	595.00	<0.001
	PHQ-9 sum	−0.00	[−0.04, 0.04]	−0.12	595.00	0.904
	OASIS sum	0.01	[−0.03, 0.06]	0.63	595.00	0.532
	LSNS sum	−0.00	[−0.03, 0.03]	0.03	595.00	0.973
	PAM13 sum	0.03	[0.00, 0.06]	2.03	595.00	0.042
	PRA sum	−0.01	[−0.01, 0.02]	0.75	595.00	0.454
	Age	−0.00	[−0.02, 0.01]	−0.69	595.00	0.492
	Sex = female	0.09	[−0.21, 0.40]	0.60	595.00	0.547
	Education = higher education	0.05	[−0.26, 0.36]	0.31	595.00	0.754
	Living situation = living with others	0.42	[0.06, 0.77]	2.28	595.00	0.023
Collective responsibility (*n* = 607)	Intercept	1.73	[1.49, 1.97]	13.52	99.11	<0.001
	PHQ-9 sum	−0.01	[−0.03, 0.02]	−0.42	593.65	0.676
	OASIS sum	−0.00	[−0.03, 0.02]	−0.09	594.71	0.930
	LSNS sum	−0.00	[−0.02, 0.01]	−0.37	594.54	0.710
	PAM13 sum	−0.01	[−0.03, 0.00]	−1.45	596.67	0.147
	PRA sum	−0.02	[−0.03, −0.01]	−4.10	596.62	<0.001
	Age	−0.01	[−0.01, 0.00]	−1.74	587.01	0.083
	Sex = female	−0.20	[−0.38, −0.02]	−2.21	596.22	0.028
	Education = higher education	−0.04	[−0.23, 0.14]	−0.41	576.64	0.679
	Living situation = living with others	−0.02	[−0.22, 0.20]	−0.15	593.60	0.885

**Table 6 vaccines-11-01795-t006:** Significant associations between patient-dependent variables and the psychological antecedents for vaccinations against influenza. Cells representing significant associations are highlighted in orange. Those associations relevant to both COVID-19 and influenza are highlighted in a dark orange. (**p* ≤ 0.05, ***: *p* ≤ 0.001).

Influenza Vaccine	Confidence	Complacency	Constraints	Calculation	Collect. Respons.
Patient-activation (PAM13)	*0.03 (*p* = 0.024)	***−0.03 (*p* = 0.001)		*0.03 (0.042)	
Doctor/patient relationship (PRA)	***0.02 (*p* < 0.001)	***−0.02 (*p* < 0.001)	*−0.01 (*p* = 0.018)		***−0.02 (*p* < 0.001)
Age	***0.02 (*p* < 0.001)	***−0.02 (*p* < 0.001)	*−0.01 (*p* = 0.024)		
Female sex					*−0.20 (*p* = 0.028)
Higher education			*−0.20 (*p* = 0.024)		
Living with others				*0.42 (*p* = 0.023)	

## Data Availability

The data generated and/or analyzed during the current study are not publicly available due to reasons of data protection but are available from the corresponding author on reasonable request. Ethics approval, participant permissions, and all other relevant approvals were granted for this data sharing.

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
