# Peer review of "Psychological Determinants of Vaccination Readiness against COVID-19 and Seasonal Influenza of the Chronically Ill in Primary Care in Germany—A Cross-Sectional Survey"

_vaccines, 2023, doi:10.3390/vaccines11121795_

Round 1

Reviewer 1 Report

Comments and Suggestions for Authors

The article by Sanftenberg et al. deals with the psychological factors of chronically ill patients associated with the compliance of vaccination against COVID-19 and seasonal influenza. Although the manuscript is very interesting, some minor issues shall be taken into account before publication. 

Throughout the manuscript:

- Please correct "Covid-19" with "COVID-19".

- Please add a space before citation number.

Abstract:

Numbers in parenthesis are not necessary. Please delete it.

Introduction:

Some more information about the safety and effectiveness of the vaccine against COVID-19 shall be added. Here is an article that might be useful: 10.1016/j.humimm.2022.08.004

Materials and methods:

Adequately described.

Result:

Full correlation and regression tables shall be inserted.

Discussion:

I suggest to not divide it into paragraph, but to make a fluent speech about it. Also, some more studies shall be added, especially about the "vaccination gate" happened during COVID-19.

Conclusion:

It is supported by the results. Adequate.

Author Response

Please see the attachment (Word file)

Reviewer 2 Report

Comments and Suggestions for Authors

Currently, there is a uncertainty  of the readiness of the population, especially patients with chronic diseases, such as depression and anxiety, regarding vaccination against COVID-19 and/or influenza. The study is interesting and extremely important because the proportion of patients with depression and anxiety is significant and the risks of complications of these two diseases in patients with comorbidities are increased. However, after reading the paper, I have some questions and comments which need to be answered.

Author Response

Please see the attachment (Word file)

Reviewer 3 Report

Comments and Suggestions for Authors

Dear authors,

This research showed important aspect of vaccination readiness among chronically ill patient, especially in COVID-19 time. 

The topic is relevant in this filed, not so much original because some similar research has been conducted. But I found it relevant and original for German population and novelty is that the authors used participants with various chronic illness. Novelty is also that authors tried to determined 5C model among this vulnerable population.

It added broader understanding of vaccination readiness.

It can be compared with chronically ill patient without psychological co-morbidity (anxiety, depression) and to compare differences between groups (ie. Cancer patient, diabetes, CHD and so on).

The conclusions are consistent with the evidence and arguments presented, and the main question is addressed.

References are appropriate and relevant to the study

Tables and figures are clearly presented, they are informative and easy to follow.

I have some minor suggestions:

Line 48 instead Covid-19, please replace with COVID-19

Line 50 – same as line 48

Line 96 – the same

Did you find/calculate differences between participants living in rural and urban areas in vaccination readness?

Round 2

Reviewer 2 Report

Comments and Suggestions for Authors

The authors satisfactorily answered all the questions and adopted most of the given suggestions, which significantly improved the revised version of the manuscript, which deserves to be published.